# Temporal and Spatial Variations in Soil Elemental Stoichiometry Coupled with Alterations in Agricultural Land Use Types in the Taihu Lake Basin

Chonggang Liu [1,2,3,†] , Jiangye Li [4,5,6,†], Wei Sun [1,2] , Yan Gao [4,5,6,*], Zhuyun Yu [7], Yue Dong [8] and Pingxing Li [1,2,*]

1   Nanjing Institute of Geography & Limnology, Chinese Academy of Sciences, Nanjing 210008, China
2   Key Laboratory of Watershed Geographic Sciences, Chinese Academy of Sciences, Nanjing 210008, China
3   University of Chinese Academy of Sciences, Beijing 100049, China
4   Institute of Agricultural Resources and Environment, Jiangsu Academy of Agricultural Sciences, Nanjing 210014, China
5   Key Laboratory of Agricultural Environment on the Lower Yangtze River Plain, Ministry of Agriculture and Rural Affairs, Nanjing 210014, China
6   National Agricultural Experimental Station for Agricultural Environment, Luhe, Ministry of Agriculture and Rural Affairs, Nanjing 210014, China
7   Hebei Province Key Laboratory of Wetland Ecology and Conservation, Hengshui University, Hengshui 053000, China
8   Scientific Research Base Administration Office, Jiangsu Academy of Agricultural Sciences, Nanjing 210014, China
*   Correspondence: ygao@jaas.ac.cn (Y.G.); pxli@niglas.ac.cn (P.L.)
†   These authors contributed equally to this manuscript.

**Abstract:** Soil elemental stoichiometry, expressed as the ratios of carbon (C), nitrogen (N), and phosphorus (P), regulates the biogeochemical processes of elements in terrestrial ecosystems. Generally, the soil C:N:P stoichiometry characteristics of agricultural ecosystems may be different from those of natural ecosystems, with distinct temporal and spatial variations along with the alterations of agricultural land use types (LUTs). The balance of soil C, N, and P reflected by their stoichiometry is primarily important to microbial activity and sustainable agricultural development. However, information on soil stoichiometric changes after long-term alterations in land use is still lacking. We characterized the temporal and spatial changes in soil elemental stoichiometry coupled with alterations in agricultural LUTs in the Taihu Lake basin. By using the ArcGIS method and meta-data analysis, our results showed that the C:N, C:P, and N:P ratios of agricultural soil in the Taihu Lake basin were much lower than the well-constrained values based on samples from forest, shrubland, and grassland at a global scale. Generally, these elemental ratios in soils increased from the 1980s to the 2000s, after experiencing changes from agricultural to other land use. The soil C:N:P stoichiometry may have maintained the increasing trend according to the meta-data analysis from the 341 peer-reviewed publications since 2010. Nevertheless, different regions showed inconsistent change patterns, with the Tianmu Mountain area surrounding the downstream of the Taihu Lake basin experiencing a reduction in those ratios. The changes in LUTs and their corresponding management practices were the major drivers shaping the spatial and temporal distributions of soil C:N, C:P, and N:P. Paddy soil generally achieved higher C sequestration potential due to more straw input and a more rapid transfer of straw C into soil C in the upstream of the Taihu Lake basin than other land use types. These results provide valuable information for the agricultural system of intensive cultivation on how their soil elemental stoichiometry characteristics vary temporally and spatially due to the alteration of agricultural land use types.

**Keywords:** temporal-spatial distributions; C:N:P stoichiometry characteristics; land use type changes; sustainable agricultural development; C sequestration



## 1. Introduction

Life in terrestrial ecosystems is supported by many macroelements, such as carbon (C), nitrogen (N), and phosphorus (P). Their concentrations, especially their stoichiometry, not only regulate the biodiversity and growth of plants and microorganisms but also determine soil organic matter (SOM) decomposition-accumulation dynamics and C stocks, which commonly support soil health [1–3]. These provide essential ecosystem functions and services, such as the promotion of sustainable food production, maintenance of soil fertility, regulation of greenhouse gas emissions, and regulation of water quality in watersheds [4]. Therefore, the soil elemental stoichiometry (e.g., C:N:P), which considers the balance of energy (C) and nutrient elements (N, P), has attracted increasing attention.

Soil total organic carbon (TOC) dynamics are regulated by elemental stoichiometry [5,6]. As the dominant constituent of SOM, TOC represents one of the most commonly determined soil attributes. The TOC/TN (total N) ratio (C:N) provides valuable information concerning the status of decomposition and potential nutrient availability in organic matter [7,8]. TN/TP (Total P) and TOC/TP have also been widely recognized and studied because N and P are the main growth-limiting nutrients for plants and microorganisms in soil [9]. The stoichiometry between C, N, and P (C:N, C:P, N:P) might regulate SOM dynamics and soil microbial carbon use efficiency (CUE) [10–13]. Currently, most studies mainly focus on the effect of different fertilization treatments on soil stoichiometry; the effects of land cover and land use, however, have rarely been mentioned [14].

Alterations in land cover and land use, driven by anthropogenic activities, are a global phenomenon that may affect the soil elemental stoichiometry [15,16]. Since the last century, the growing demand for food has led to intensive agriculture in China due to the surge in populations, and large numbers of forests or grasslands have been replaced by agricultural land. Vegetation have been drastically modified as a result, and soils have been exposed to radical changes, mainly in plant diversity, water balance, and C and nutrient dynamics [17–20]. Moreover, agronomic practices involve many mechanical and chemical disturbances that modify soil chemistry, physics, and its microbial communities [21–23]. It is well documented that land-use changes for various agricultural purposes have resulted in losses of 25~50% of soil organic carbon (SOC) in soil originating from forests [24,25], shrublands [26], and grasslands [27]. Moreover, modern agriculture, with its imbalance in the use of N and P fertilizer, may create an unprecedented human-induced imbalance of C:N:P ratios in agricultural soil [28,29]. The combination of these factors could generate dramatic stoichiometric shifts in soils due to land-use change (LUC). In addition, different crops vary in the demand for fertilizer and management practices, which are bound to induce the disparity of soil C:N:P stoichiometry and their C storage capacities between different croplands [30,31]. However, the temporal and spatial changes in elemental stoichiometry in agricultural soil and their relations with LUC remain unknown.

The Taihu Lake basin provides an excellent opportunity to study the temporal and spatial changes in elemental stoichiometry in agricultural soil and their relations with LUT changes due to its unique environmental and climatic characteristics and LUT changes resulted from fast-growing socioeconomic conditions in China. The land surface in this region undergoes annual disturbance and change through urbanization, logging of forestry to grow trees of economic value, changing cropland to orchard or vegetable land, and farming practices such as crop rotation and ploughing [32,33]. This region has been regarded as one of the major grain-producing areas in China. Its warm weather and sufficient water supply allow the growth of two crops a year, e.g., double-cropping of rice or rice–wheat rotation. However, the agricultural land use types in this region are experiencing unprecedented change due to the aim of higher profits. In 1985, the rice and wheat planting area in this region was approximately $171.8 \times 10^4$ hm² [34,35], while the main crop planting area ($115.5 \times 10^4$ hm²) was reduced by 32.8% in 2010 [36]. Much SOC is at risk of being transferred to the atmosphere via deforestation, land clearing, and other forms of land use type alteration driven largely by an increasing population [36,37]. Moreover, the unbalanced use of N and P fertilizers may aggravate the imbalance of

soil C:N:P stoichiometry. As a result, this may lead to alterations in ecosystem function and services.

In this study, our major objectives were (1) to characterize the temporal and spatial changes in topsoil C, N, and P concentrations and their ratios, and (2) to further investigate the correlations between the concentration of C, N, or P and the corresponding C:N or C:P to reflect whether the ecological stoichiometry were still able to regulate the stocks of soil C, N, or P, with alterations in land use types in the Taihu Lake basin from the 1980s to the 2010s. In the present study, one hypothesis was tested: the soil C:N, C:P, and N:P ratios increased with time, especially in paddy fields. This research is critically important to learn how land-use changes impact ecosystem services, and would provide practical significance to proper land use for sustainable agricultural development.

## 2. Material and Methods

### 2.1. Study Area

The Taihu Lake basin (Figure 1A), located on the east coast of China (119°3′1″~121°54′26″ E, 30°7′19″~32°14′56″ N), is one of the most developed regions of China and it was where the land use types experienced a dramatic change from 1980 to 2016 with the development of the economy (Figure 1B). Taihu Lake (2238.1 hm²) is the largest lake in this region and the third largest freshwater lake in China. The basin has a typical subtropical monsoon climate, with an annual mean temperature of 15~17 °C and annual mean precipitation of 1010~4000 mm. The dominant soil types were yellow brown soil and paddy soil.

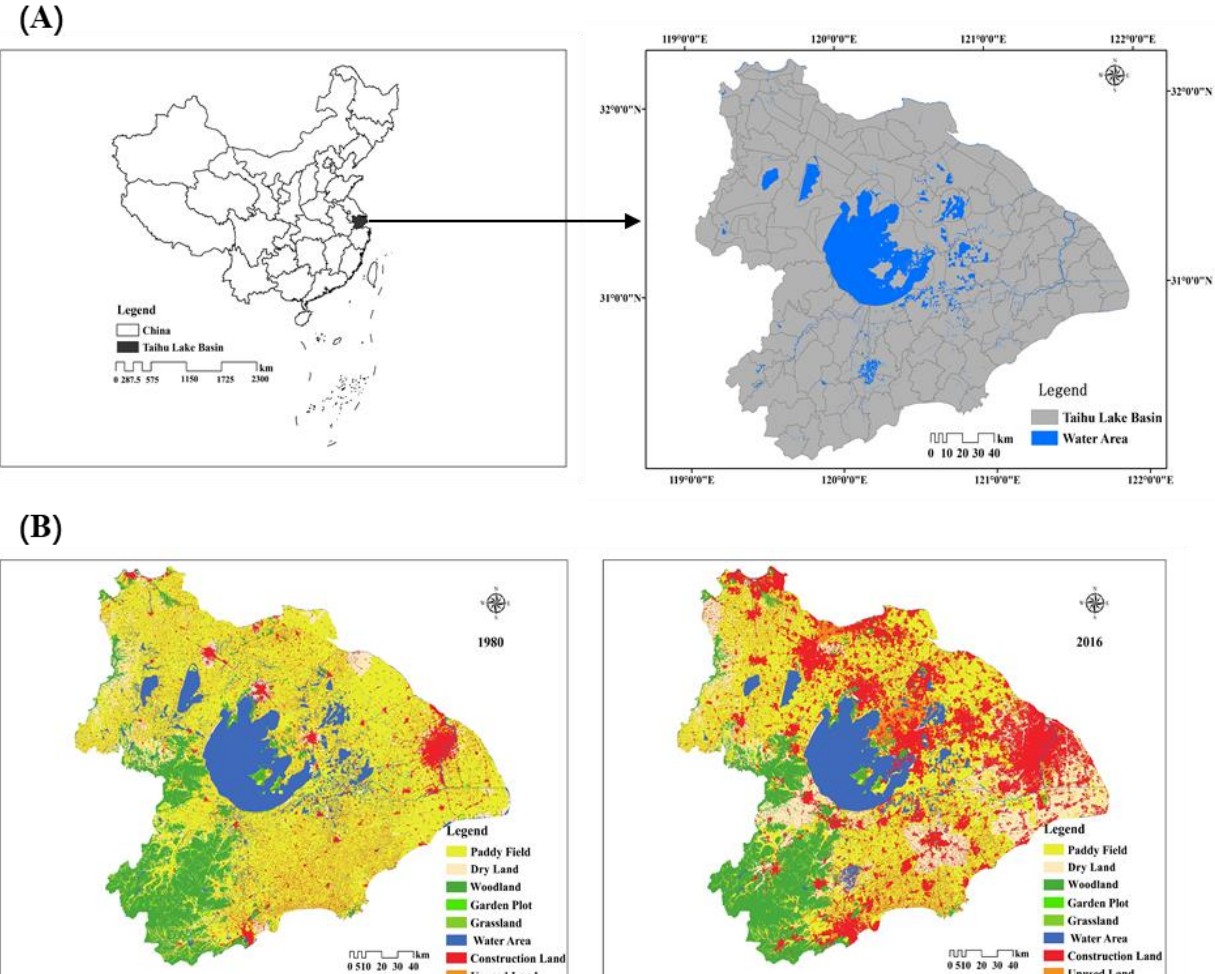

**Figure 1.** The site of the Taihu Lake basin (**A**) and land use type changes from 1980s to 2000s in Taihu Lake basin (**B**).

Generally, there was a net increase in woodland area for these two periods (11.42 hm$^2$ from 1980 to 2000; 137.72 hm$^2$ from 2000 to 2016), which was mainly converted from dry land during 1980–2000 and from paddy fields, dry land, and grassland during 2000–2016 (Table S1). However, there was a net decrease in grassland area for these two periods (3.8 hm$^2$ from 1980 to 2000; 81.18 hm$^2$ from 2000 to 2016), which was mainly converted to construction land during 1980–2000 which was then converted to woodland during 2000–2016.

### 2.2. Data Collection

We collected soil TN, TP, and SOC data for Taihu Lake basin in 1980 from the second Chinese Soil Survey, and the data for the year 2000 from Soil Series Survey for Jiangsu and Zhejiang Province. These two surveys used the same sampling sites. There was a total of 1451 data for each parameter of TN, TP, and SOC in each survey. In this study, only the surface (0 to 20 cm) soil data were used as it directly recorded the impact of land-use changes in the time scale of decades. We collected the data after 2010 from peer-reviewed papers published in both Chinese and English journals with regard to major ALUTs in Taihu Lake region, e.g., one season rice, rice–wheat rotation, orchard, tea plantation. Studies in Chinese were collected from the China National Knowledge Infrastructure (CNKI) database and those in English from ISI-Web of Science. The key words used for searching were soil organic matter, soil organic carbon, total nitrogen, total phosphorus, paddy field, wheat, orchard (e.g., grape, peach, pear), tea plantation. Papers published were only included if they met the following criteria: (1) studies had to report at least two variables including soil organic matter/soil organic carbon, total nitrogen, total phosphorus in order to calculate C:N, C:P, and N:P stoichiometry; (2) any studies lacking replication were not considered; (3) only the experimental data obtained between 2015 and 2020 were considered to represent more recent soil characteristics. A total of 341 peer-reviewed publications (204 observations for paddy field, 62 observations for rice–wheat rotation, 46 observations for Orchard, 29 observations for tea plantation) were selected for our analysis.

Among them, a total of 108 samples were measured in laboratory in 2018. TOC, TN, and TP contents were determined by a CHNS elemental analyser (Carlo Erba model EA1108, Italy Vario). The total P (TP) content in the soil was measured using perchloric acid digestion [38]. Additionally, the soil stoichiometry was calculated by the ratio of TOC to TN (C:N), TOC to TP (C:P), and TN to TP (N:P).

### 2.3. Data Processing and Calculation

2.3.1. Mapping Method for Spatial Distribution and Variation of Soil C:N, C:P, and N:P in 1980~2000

Based on the 1451 data that we obtained from China's second soil survey (1980) and the Soil Series Survey for Jiangsu and Zhejiang Province in 2000, the 1451 sampling points were generated according to the latitude and longitude in ArcGIS 10.8. The mass ratio of C:N, C:P, and N:P was calculated based on the concentrations of soil TOC, TP, and TN. The changing rates for these parameters between 1980 and 2000 were calculated according to their differences in 2000 and 1980. The Kriging method was used to interpolate and obtain the spatial distribution of C:N, C:P, N:P, and their changing rates in the Taihu Lake basin, respectively. The Kriging method weighed the area between multiple sampling points to derive an estimate of the unmeasured position. This method is similar to the inverse distance weighting method, which consists of a weighted sum of the data:

$$\hat{Z}(s_0) = \sum_{i=1}^{N} \lambda_i Z(s_i) \tag{1}$$

where $Z(s_i)$ represents the measured value at the position of the $i$-th sample point, $\lambda_i$ represents the unknown weight of the measured value at the position of the $i$-th sample point, and the weight $\lambda_i$ depends on the distance between the sample point, the predicted position, and the predicted position A fitted model of the spatial relationship between the

measured values of the sample points; $s_0$ represents the predicted position; $N$ represents the measured value of the sampled point.

The "nature break" method was selected to cluster the spatial interpolation. These classes were based on natural groupings inherent in the data. ArcMap 10.8 identified break points by picking the class breaks that best grouped similar values and maximized the differences between classes. The features were divided into classes whose boundaries were set where there were relatively big jumps in the data values.

### 2.3.2. Computation Method of Proportion of Different Land Use Types in Each Index Interval

The changes of C:N, C:P, and N:P between 1980 and 2000 were intersected with land-use changes in 1980 and 2000, respectively. The intersected layers were recalculated while different land use types and their areas in 1980 and 2000 were extracted according to the divided intervals as described in Section 2.3.1. Through the comparison of the land use data between the two periods, we could obtain the main land use type changes in different C:N, C:P, and N:P changing intervals, and then analyze the correlations between them. These classes were based on natural groupings inherent in the data. ArcMap identified break points by picking the class breaks that best grouped similar values and maximized the differences between classes. The features were divided into classes whose boundaries were set where there were relatively big jumps in the data values.

### 2.3.3. Statistical Analysis

The ratios of soil C:N, C:P, and N:P in 1980 and 2000 in the Taihu Lake basin were also shown in box plots conducted by Origin 9.1. The differences of C:N, C:P, and N:P between 1980 and 2000 were measured by paired-sample *t*-test in SPSS 20.0 with a 95% confidence interval. Linear regression analysis was conducted after the Pearson product-moment correlation analysis by two-tailed test in SPSS 20.0. Correlations between topsoil C:N, C:P, or N:P and corresponding C, N, or P concentrations were conducted in Excel 2016 and the statistical analysis was conducted using a *t*-test in SPSS 20.0.

## 3. Results

### 3.1. Temporal and Spacial Changes in Soil C:N:P Stoichiometry in the Taihu Lake Basin

In the 1980s (Figure 2A), the soil C:N in most areas of the Taihu Lake basin ranged from 9 to 10 and 10 to eleven, accounting for 42.6% and 26.5% of the total area, respectively. A total of 19.8% of the area showed a soil C:N ratio lower than 9, while only 11.1% of the area showed a soil C:N ratio higher than 11. In the 2000s (Figure 2B), the area with soil C:N ranging from 10 to 11 greatly increased when compared to the 1980s, with 42.4% of the area showing soil C:N ranging from 10 to 11. Only 4.76% of the area presented a soil C:N ratio lower than 9, while 15.61% of the area showed a C:N ratio higher than 11.

In the 1980s (Figure 2C), the soil with C:P lower than 28 accounted for 61.88% of the total area (20.9% of the total area had C:P < 18; 27.3% of the total area had C:P ranging from 18 to 24; 12.4% of the total area had C:P ranging from 24 to 28). In the 2000s (Figure 2D), the soil C:P ratio in the Taihu Lake basin was generally enhanced, with the area of soil C:P > 28 accounting for 67.0% of the total area.

In the 1980s (Figure 2E), the soil N:P ratio in the Taihu Lake basin was mainly distributed in the ranges of ≤2, two~three, and 2~4, which accounted for 24.7%, 38.1%, and 28.0% of the total area, respectively. Soil with N:P ≥ 4 accounted for only 9.3% of the total area in the Taihu Lake basin. In the 2000s (Figure 2F), the area with soil N:P ranging from 2 to 3, 3 to 4 and ≥4 increased to 46.0%, 30.9%, and 13.1% of the total area, respectively, while the area with soil N:P ≤ 2 decreased to 10.1% of the total area.

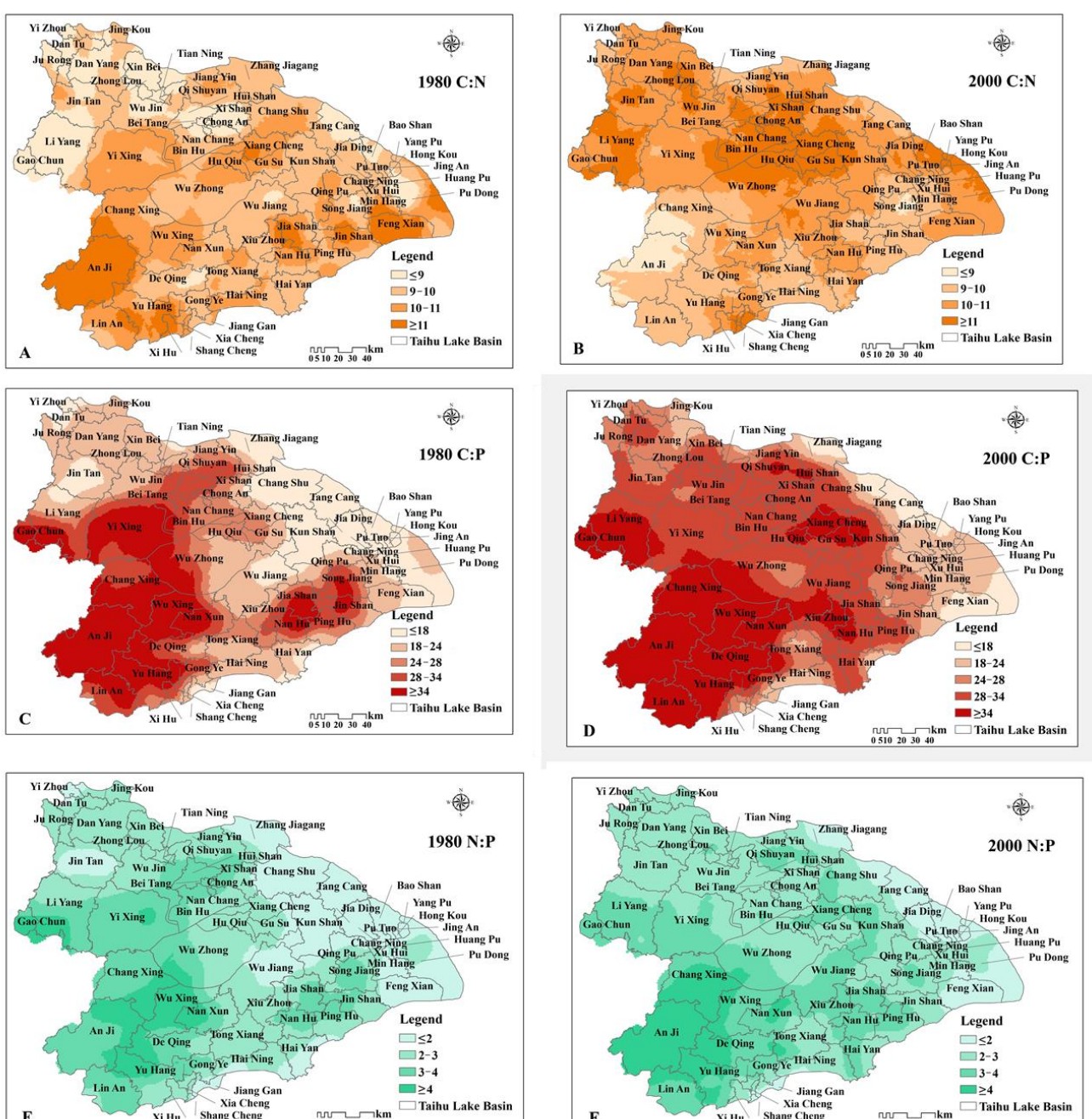

**Figure 2.** Temporal and spatial distribution of topsoil C:N, C:P, and N:P in 1980s (**A**,**C**,**E**) and 2000s (**B**,**D**,**F**), respectively, in Taihu Lake basin.

According to Figure 2 and relevant calculations, in the 1980s, the soil C:N ranged from 9 to 10 was mainly from paddy fields (87.8%), while the soil C:N ranged from 10 to 11 was from both some paddy fields (59.2%) and woodlands (32.2%). Most of these areas were evenly distributed in the central Taihu Lake basin. A total of 19.8% of the area showed a soil C:N ratio lower than 9 and was mainly scattered in the northwest Taihu Lake basin. Only 11.08% of the area showed a soil C:N ratio higher than 11 and was mainly scattered in the region surrounding the southwestern Taihu Lake basin, especially in the mountainous area (44.8%) in Anji and Lin'an of Zhejiang Province.

In the 2000s, the area with soil C:N ratios ranging from 10~11 was mainly distributed in paddy fields (85.7%) between Nanjing and Shanghai located in the northern Taihu Lake basin. A total of 15.6% of the total area showed C:N higher than 11 distributed in Wuxi and

Changzhou, where both industry and agriculture are well developed. In contrast, 4.8% of the area with soil C:N lower than nine was mainly distributed in the mountainous area of Anji and Lin'an of Zhejiang Province, which used to have high soil C:N in the 1980s.

In the 1980s, the soil with a C:P ratio lower than 28 was mainly distributed in paddy fields (86.9~88.8%) located in the region surrounding the downstream Taihu Lake basin along the Yangtze River. The soil with a C:P higher than 28 was mainly distributed in woodland areas and paddy fields located in the mountainous regions surrounding the upstream of the Taihu Lake basin.

In the 2000s, the soil C:P ratio in the Taihu Lake basin was generally enhanced, especially in the region surrounding the downstream of the Taihu Lake basin along the Yangtze River, in some areas surrounding large- and medium-sized cities (e.g., Suzhou and Wuxi), which used to have low C:P in the 1980s. The area with soil C:P ranging from 28 to 34 was mainly distributed in the paddy field (69.8%), while soil with C:P > 34 was mainly distributed in both the mountainous area (41.1%) and some paddy fields (50.9%).

In the 1980s, relatively high soil N:P, e.g., N:P ranged from three to four and ≥four, was mainly distributed in the woodland area and some paddy fields located in the region surrounding the downstream Taihu Lake basin along the Yangtze River. Relatively lower soil N:P, e.g., N:P ranged from 2 to 3 and ≥2, was mainly distributed in some paddy fields located in the mountainous regions surrounding the upstream Taihu Lake basin. In the 2000s, higher soil N:P was also mainly distributed in woodlands and some paddy fields, while lower soil N:P was mainly distributed in some paddy fields.

The box plots showed that the median values of soil C:N, C:P, and N:P in the 2000s were consistently higher than those observed in the 1980s (Figure 3, $p < 0.05$). The mean C:N, C:P, and N:P mass ratios of the surface layer (0~20 cm) soil in the Taihu Lake basin were 9.6, 21.9, and 2.3 in the 1980s and 10.4, 30.1, and 2.8 in the 2000s, respectively, and the differences in the same elemental stoichiometry between the 1980s and 2000s were all significant ($p < 0.05$).

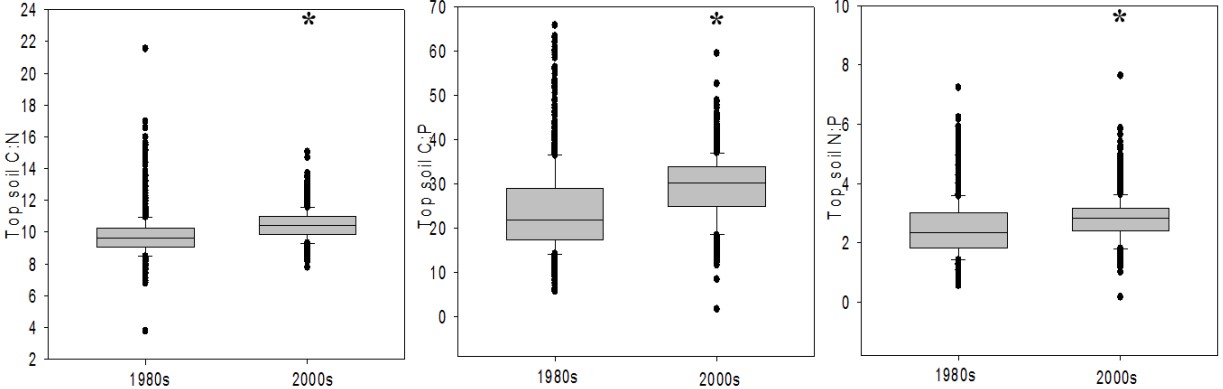

**Figure 3.** The vertical box plot of topsoil C:N, C:P, and N:P ratios in 1980s and 2000s, respectively, in Taihu Lake basin. The boundary of the box closest to zero indicated the 25th percentile, a solid line within the box marked the median, and the boundary of the box farthest from zero indicated the 75th percentile. Whiskers (error bars) above and below the box indicated the 90th and 10th percentiles. "*" indicates the difference was significant at 5% level.

### 3.2. Alteration of Soil C:N, C:P and N:P Coupled with Agricultural Land Use Type Changes between 1980 and 2000

From the 1980s to the 2000s, paddy fields suffered dramatic changes, accounting for 80.8% of the total altered agricultural land area (the sum of paddy field, dryland, woodland, grassland, and unused land). Nevertheless, paddy fields were still the major LUTs in this region, consisting of 75.5% of the land area with the LUTs remained unchanged (Table 1). Among the unchanged paddy fields, 69.0% of the fields experienced a substantial increase in soil C:N (21.0% of paddy had an increase in C:N < 10%; 24.9% of paddy fields had

an increase in C:N ranging from 10 to 20%; 15.9% of paddy fields had an increase in C:N > 20%), whereas 31.0% of paddy fields experienced an obvious reduction in soil C:N (21.6% of paddy fields had a decrease in C:N < 10%; 9.3% of paddy fields had a decrease in C:N > 10%). Although the paddy fields were mainly occupied by construction land, they were transformed to drylands and woodlands. Most of these agricultural soils (85.2%) alternated from paddy fields showed an obvious increase in soil C:N (25.1% of paddy fields had an increase in C:N <10%; 36.2% of paddy fields had an increase in C:N ranging from 10 to 20%; 23.9% of paddy fields had an increase in C:N >20%), whereas 14.8% of paddy fields experienced an obvious reduction in soil C:N (13.4% of paddy fields had a decrease in C:N < 10%; 1.3% of paddy fields had a decrease in C:N > 10%).

**Table 1.** Alteration of soil C:N, C:P, and N:P coupled with agricultural land use types changes between 1980 and 2000.

| | | Unchanged (hm$^2$) | | | | | Changed (hm$^2$) | | | | |
|---|---|---|---|---|---|---|---|---|---|---|---|
| | | Paddy Field | Dry Land | Woodland | Grassland | Unused Land | Paddy Field | Upland Field | Woodland | Grassland | Unused Land |
| Changes of soil C: N (%). | ≤−20 | 77,853.79 | 10,942.83 | 147,429.43 | 5874.96 | 16.65 | 620.08 | 1656.62 | 428.58 | 0.00 | 12.43 |
| | −20–−10 | 142,694.02 | 9422.41 | 108,912.23 | 1410.83 | 35.32 | 1796.95 | 1526.09 | 206.57 | 0.00 | 4.04 |
| | −10–0 | 513,178.59 | 22,414.50 | 164,748.59 | 4850.84 | 166.26 | 23,873.02 | 5031.70 | 983.25 | 374.41 | 30.41 |
| | 0–10 | 668,724.27 | 49,357.48 | 80,318.71 | 2249.48 | 425.65 | 44,534.01 | 8460.33 | 1794.84 | 146.88 | 69.92 |
| | 10–20 | 590,317.97 | 38,555.91 | 35,746.25 | 1130.68 | 623.65 | 64,177.17 | 9330.64 | 1363.75 | 99.41 | 157.47 |
| | ≥20 | 377,667.73 | 57,929.54 | 24,923.92 | 1208.59 | 210.26 | 42,502.03 | 10,185.77 | 279.68 | 4.59 | 0.00 |
| Changes of soil C: P (%) | ≤−20 | 146,587.00 | 9108.14 | 26,910.83 | 1097.35 | 128.76 | 6902.43 | 1171.05 | 294.70 | 0.00 | 17.67 |
| | −20–0 | 438,790.96 | 30,831.96 | 196,993.76 | 3966.87 | 27.58 | 17,470.28 | 4686.35 | 656.82 | 368.63 | 20.93 |
| | 0–20 | 528,502.10 | 40,372.40 | 199,546.41 | 6205.09 | 744.01 | 38,351.01 | 8139.33 | 1192.67 | 101.40 | 42.95 |
| | 20–40 | 494,907.77 | 46,879.72 | 107,773.23 | 4412.91 | 341.55 | 53,162.51 | 11,567.71 | 2013.63 | 124.78 | 0.00 |
| | 40–60 | 270,568.22 | 22,323.22 | 20,865.31 | 544.00 | 183.68 | 21,384.68 | 5624.71 | 581.01 | 5.29 | 157.43 |
| | ≥60 | 491,080.22 | 39,106.73 | 9989.76 | 499.14 | 52.19 | 40,232.36 | 5001.80 | 317.86 | 25.18 | 35.30 |
| Changes of soil N: P (%) | ≤−20 | 81,472.90 | 9884.22 | 17,319.89 | 1310.53 | 131.83 | 5881.10 | 1017.75 | 101.06 | 11.12 | 23.29 |
| | −20–0 | 594,164.88 | 44,217.47 | 83,620.59 | 2435.38 | 257.54 | 39,964.58 | 8941.48 | 708.17 | 145.79 | 8.82 |
| | 0–20 | 670,129.20 | 57,037.72 | 214,093.68 | 3548.60 | 616.70 | 60,134.29 | 12,003.25 | 1742.88 | 313.09 | 26.30 |
| | 20–40 | 459,801.07 | 37,325.82 | 108,106.22 | 3976.50 | 354.78 | 32,614.62 | 7035.71 | 1818.24 | 130.10 | 174.64 |
| | 40–60 | 228,020.72 | 20,451.19 | 55,808.84 | 2713.29 | 100.61 | 9792.92 | 2368.16 | 318.42 | 0.00 | 0.00 |
| | ≥60 | 336,847.79 | 19,706.46 | 83,129.97 | 2741.06 | 16.65 | 29,115.76 | 4824.60 | 367.90 | 25.18 | 41.24 |

Except for the construction land, a large area of dry land was transformed to paddy fields and woodlands. A total of 77.3% of the altered dry land had a dramatic increase in soil C:N, e.g., 23.4% of dry land showed an increase of <10%; 25.8% of dry land showed an increase in C:N ranging from 10 to 20%; and 28.1% of dry land showed an increase in C:N > 20%. Meanwhile, 22.7% of the alternating dry land experienced a decrease in soil C:N. The dry land that remained unchanged had a similar increasing trend for soil C:N, with 77.3% of the unchanged dry land showing a substantial increase in soil C:N, but 22.7% of the unchanged dry land showing an obvious decrease in soil C:N.

The woodland area generally experienced little change. However, there was a contrasting trend for soil C:N changes between woodland areas that underwent alteration and remained unchanged. Among woodlands remaining unchanged, 74.9% of woodlands soil experienced an obvious decrease in soil C:N, with 29.3% woodland area showing a decrease in C:N < 10%; 19.4% woodland area showing a decrease in C:N ranging from 10 to 20%; and 26.2% woodland area showing a decrease in C:N > 20%. In contrast, among woodlands that has been transformed to other LUTs, 68.0% of woodland soil had an obvious increase in soil C:N, with 35.5% woodland area showing an increase in C:N < 10%; 27.0% woodland area showing an increase in C:N ranging from 10 to 20%; and 5.5% woodland area showing an increase in C:N > 20%.

Most paddy fields, drylands and woodlands, regardless of their LUT changes, experienced increases in soil C:P and N:P. A total of 75.3% and 71.5% of the paddy fields that remained unchanged increased in soil C:P and N:P, respectively. Meanwhile, 86.3% and 74.2% of the altered paddy field experienced a substantial increase in soil C:P and N:P as well, respectively. In addition, 78.8% and 71.3% of the dry land that remained unchanged

had increases in soil C:P and N:P, respectively. Similarly, 83.8% and 72.5% of the alternated dry land experienced a substantial increase in soil C:P and N:P, respectively.

*3.3. Responses of Soil Elemental Stoichiometry to the Changes in the C, N, and P Pools in Soil between the 1980s and 2000s*

In the 1980s, topsoil C:N was significantly positively correlated with soil TOC, whereas it had no significant correlation with soil TN (Figure 4). In contrast, in the 2000s, topsoil C:N did not show a significant correlation with soil TOC, but it was significantly negatively correlated with soil TN. The topsoil C:P in the 1980s and 2000s was significantly positively correlated with soil TOC, while it was significantly negatively correlated with soil TP. The topsoil N:P in the 1980s and 2000s were all negatively correlated with soil TP, while they showed a positive correlation with soil TN in the 1980s but a negative correlation with soil TN in the 2000s. Extremely similar correlation patterns between topsoil N:P or C:P and TP were observed for both the 1980s and 2000s, showing a decrease in N:P or C:P accompanied by an increase in soil TP concentrations.

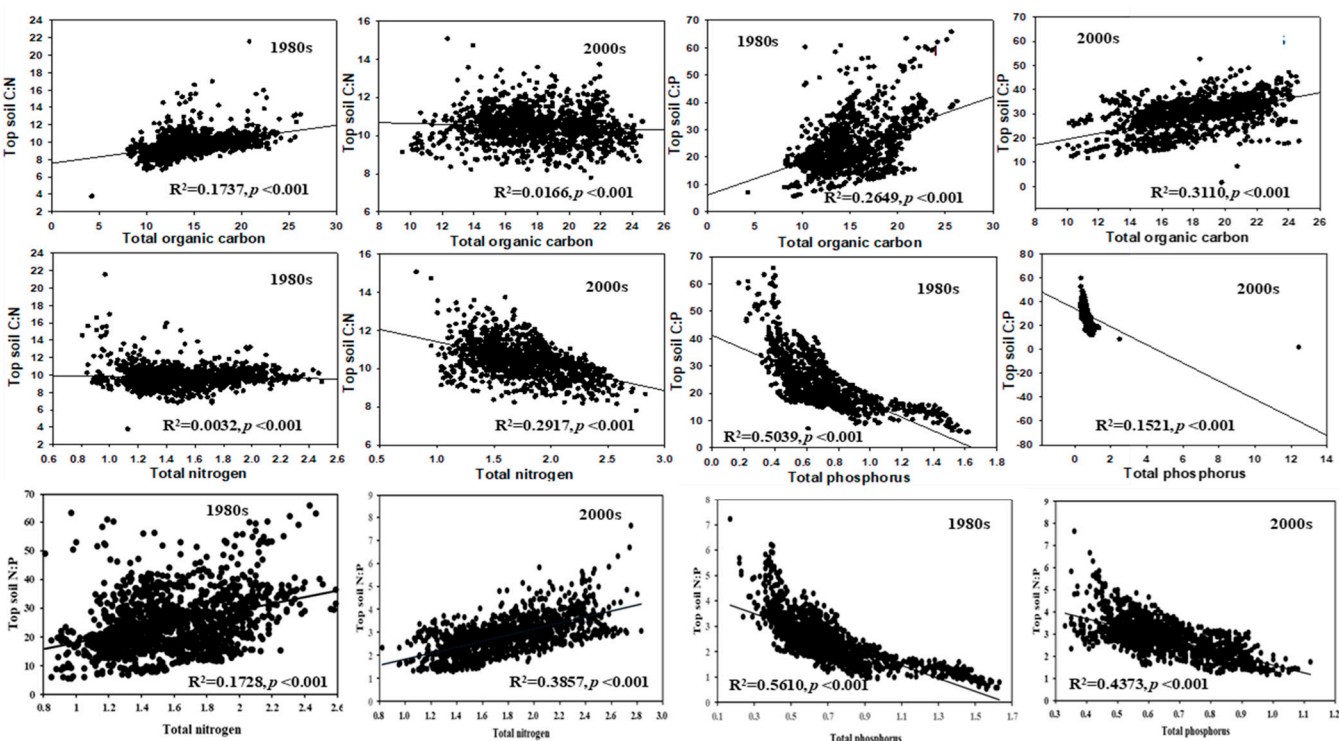

**Figure 4.** Correlations between topsoil C:N, C:P, or N:P and corresponding C, N, or P concentrations.

*3.4. Meta-Analysis of Soil C:N, C:P, and N:P under Different Land Use Types after 2000s*

The topsoil C:N, C:P, and N:P ratios under the major ALUTs (paddy fields, paddy-wheat rotation, orchard) in the Taihu Lake basin during different periods (1980s, 2000s, 2015–2020) are shown in Table 2 according to data collected from published literature. Overall, the median values of C:P for rice fields, fields with rice–wheat rotations, and orchards increased from the 1980s and 2000s to the present (2015~2018), which was consistent with the data obtained from the soil survey, as shown in the previous section. The median values of C:N for rice fields also showed an increasing trend from the 1980s to the present (2015~2018). The median values of C:P derived from the published papers were higher than the values obtained from the soil survey for the 1980s and 2000s.

**Table 2.** Topsoil C:N, C:P, and N:P ratio under different agricultural land use types (ALUTs) in Taihu Lake basin according to meta-data analysis.

| Land Use Type | Data Number | C:N | | | C:P | | | N:P | | |
|---|---|---|---|---|---|---|---|---|---|---|
| | | Mean | Mid-Value | Min-Max | Mean | Mid-Value | Min-Max | Mean | Mid-Value | Min-Max |
| 1980s | | | | | | | | | | |
| Rice-Field | 25 | 9 | 10 | 5–11 | 57 | 49 | 14–106 | 3 | 4 | 1~6 |
| Rice–Wheat Rotation | 6 | 12 | 13 | 10–13 | 41 | 40 | 34–48 | 3 | 3 | 2~3 |
| Orchard | 3 | 14 | 14 | 13–17 | 55 | 54 | 48–63 | 4 | 4 | 3~4 |
| Tea Plantation | 4 | 5 | 5 | 5–6 | 24 | 23 | 17–31 | 5 | 5 | 3~6 |
| 2000s | | | | | | | | | | |
| Rice-Field | 97 | 11 | 10 | 6–35 | 53 | 48 | 13–103 | 5 | 5 | 1~13 |
| Rice–Wheat Rotation | 26 | 11 | 10 | 9–19 | 31 | 26 | 13–49 | 3 | 3 | 1~5 |
| Orchard | 17 | 15 | 15 | 8–26 | 45 | 48 | 17–59 | 3 | 3 | 1~6 |
| Tea Plantation | 10 | 11 | 11 | 10–13 | 38 | 31 | 19–94 | 3 | 3 | 2~7 |
| 2015–2020 | | | | | | | | | | |
| Rice-Field | 26 | 13 | 10 | 6–38 | 50 | 43 | 13–101 | 5 | 4 | 5~12 |
| Rice–Wheat Rotation | 14 | 10 | 10 | 6–13 | 41 | 42 | 22–56 | 4 | 4 | 3~6 |
| Orchard | 11 | 12 | 9 | 9–23 | 25 | 23 | 7–60 | 2 | 2 | 1~4 |
| Tea Plantation | 6 | 5 | 5 | 4–7 | 45 | 45 | 19–57 | 8 | 8 | 4~13 |

## 4. Discussion

### 4.1. The General Soil C:N:P Stoichiometry Characteristics in Taihu Lake Basin

The stoichiometry of C, N, and P is used to reveal the transitions in nutrient limitations for microorganisms and plants. Moreover, soil C:N:P stoichiometry characteristics also reflect the soil health condition. The soil C:N ratio has been widely recognized as a good indicator of the degree of SOC decomposition and accumulation processes. A previous study showed lower and less variable mass C:N ratios ranging from 9.8 to 12.4 in the 0~20 cm depth, compared with our results which showed that the mass C:N mean value was 9.7 in the 1980s; the C:N mean value was 10.7 in the 2000s; C:N mean value was 13 for paddy field and 10 for the field with rice–wheat rotation from 2015 to 2020 in the Taihu Lake basin [39].

In a recent global meta-data analysis, Cleveland and Liptzin [10] stated a remarkably constrained soil C:N:P ratio of 186:13:1 (molar ratio) on the global scale. This corresponded to soil C:N, C:P, and N:P molar ratios of approximately 14.3, 186, and 13, respectively. However, their analysis was mainly based on samples from forest, shrubland, and grassland. Our results revealed that the soil mean C:N, C:P, and N:P molar ratios in the Taihu Lake basin were 11.4, 61.8, and 5.4 in the 1980s and 12.5, 74.7, and 6.0 in the 2000s, respectively. Generally, the soil mean molar C:N ratio in the Taihu Lake basin was slightly lower than the above constrained values, while the soil mean C:P and N:P molar ratios were far lower than the constrained values. The major reason for these differences was probably that a large amount of C was removed after harvest in the agricultural field, causing a reduction in plant C input such as above litter, rhizodeposits, and root mass [40]. Afterwards, a nationwide meta-analysis for soil elemental stoichiometry across various soil groups in China was conducted by Tian et al. based on the second Chinese soil survey (National Soil Survey Office 1993, 1994a,b, 1995a,b, 1996) [40]. From the frequency distribution of soil C, N, and P ratios, they found that most C:N, C:P, and N:P ratios were in the range of 6~12, 24~48, and 3~6, respectively. The number-weighted mean soil C:N, C:P, and N:P ratios were 11.9, 61, and 5.2, respectively, which were similar to the area-weighted means (12.1, 61, and 5.0, respectively). Overall, our C:N:P stoichiometry values fell in the above ranges. Tian et al. [41] also revealed that the C:N, C:P, and N:P molar ratios of the surface organic-rich layer (0 cm~10 cm of A horizon) in China were 14.4, 136.0, and 9.3,

respectively. This finding was similar to the well-constrained values observed by Cleveland and Liptzin [10]. However, even in the mountainous area of the Taihu Lake basin, which was believed to have high C:N ratios, the soil mean C:N, C:P, and N:P ratios were lower than the well-constrained values (soil C:N:P ratio of 186:13:1). Typically, a C:N ratio above 12~14 is considered indicative of a shortage of nitrogen in the soil. The C:N ratios between nine and twelve reflect a lower degree of decomposition of the organic materials present in the Taihu Lake basin [42].

*4.2. Temporal Changes in Soil Elemental Stoichiometry in the Taihu Lake Basin*

From the 1980s to the 2000s, the soil C:N, C:P, and N:P ratios generally experienced consistent increases in the Taihu Lake basin, although few specific areas experienced obvious reductions in these ratios (Figure 2). The increase in the soil C:P and N:P ratios was supposed to be due to a more dramatic increase in soil TOC by 18.9 and TN by 13.5% with slow-growing TP by 7.5% from the 1980s to the 2000s in the Taihu Lake basin (Figure S1). According to the Statistics Yearbook of China, the amount of P fertilizer consumption in China increased by approximately 1.5 times from 2.73 million tons in 1980 to 6.91 million tons in 2000, N fertilizer consumption increased by 1.3 times from 9.34 million tons in 1980 to 21.82 million tons in 2000, while compound fertilizer with a ratio of $N:P_2O_5$ ranging from two to five increased by 31 times from 0.27 million tons in 1980 to 8.64 million tons in 2000, which may have induced the increases in both soil TN and TP content, and a greater increase in TN content than TP [43]. N addition could reduce C mineralization rates [44], which indicated the increase in N fertilizer application rate in favor of soil C storage. Although there were major missing N or P fertilizer consumption data for Jiangsu and Zhejiang Provinces in the Taihu Lake basin before 1990, relevant research revealed that the consumption of N fertilizers was higher than that of P fertilizers during 1980 and 2000 [45]. In addition, the correlation analysis revealed that soil C:P ratios were regulated by both soil TOC and TP. On the one hand, the increase in TOC in the soil was supposed to be closely related to the straw return practice. Since 1990, the Chinese government has forced the return of crop straw to the field nationwide. From 1980 to 2000, the yields of rice, wheat, and corn increased approximately 34%, 80%, and 69%, respectively [34]. Increasing primary productivity involves the increased input of organic C into the soil, especially in soils with rice–wheat rotations in southern China [43,45,46]. It was reported that the C sequestration potential of paddy soils had reached 10,148 Tg C since straw return was implemented in subtropical areas of China [34,47]. Meanwhile, organic fertilizer substituting for partial chemical fertilizers and other agronomic measures that are beneficial for C sequestration were encouraged in that time [48,49], which would further enhance soil C sequestration. Moreover, chemical fertilizer application rates have been required to be reduced in the most recent 10 years and will be kept in the reduced rate in the future. This resulted in the enhanced soil C:N (10.34 in 2000s compared with 9.79 in 1980s) and C:P ratios (29.12 in 2000s compared with 24.38 in 1980s).

Similarly, the increase in soil C:N ratios was believed to be due to a more dramatic increase in soil TOC relative to a slower increase in TN. In the 1980s, the soil C:N ratios were mainly regulated by the decomposition of soil organic matter, indicating significant positive correlations with soil TOC content. However, in the 2000s, such a correlation disappeared when the soil C:N ratios in the Taihu Lake basin experienced a slight increase as a whole. This may indicate decoupled cycles of soil C and N in the Taihu Lake basin due to the exogenous input of inorganic N fertilizers as well as organic materials, e.g., straw. According to the meta-data analysis from the published paper, the soil C:N and N:P ratios may maintain an increasing trend in most regions of the Taihu Lake basin [41].

*4.3. Spatial Changes of Soil Elemental Stoichiometry in Taihu Lake Basin*

The Taihu Lake basin presented obvious spatial differences in soil C:N:P stoichiometry characteristics. Generally, the region surrounding the downstream Taihu Lake basin experienced a great reduction in soil C:N, C:P, and N:P between the 1980s and 2000s, especially

the Tianmu Mountain area located in northeast Zhejiang Province, e.g., Anji and Lin'an city. The sharp reduction in soil C:N mainly resulted from the dramatic increase in exogenous N input between 1980 and 2000 due to the development of economic forests (e.g., walnut trees, Chinese chestnut trees, tea-oil trees, and bamboo forests) and tea plantations in those regions. Although there was a major lack of statistical data for the tea plantation area before 2010, the data thereafter clearly showed an increase in the tea plantation area in Zhejiang Province from 13,567 ha in 2011 to 198,524 ha in 2017. Such enhanced forestry output implied the combination of the increasing economic forest area and the production that is accompanied by increased N fertilizer input. The N fertilizer amount used in northeastern Zhejiang Province (total is 24.81 tons in 2017), where Anji, Lin'an, and Jiaxing (Tianmu Mountain area) are located, was higher than that in southwestern Zhejiang Province (the total was 18.09 tons in 2017). At the same time, this region (e.g., surrounding Lin'an city) experienced a slight decrease in soil TOC content. As a result, the soil C:N ratios in the region underwent a dramatic decrease. Similarly, the decrease in soil C:P in some regions of the Tianmu Mountain area was also mainly due to the increase in P input but a slight decrease in soil TOC. The increase in soil TN was greater than that in soil TP, resulting in the increase in soil N:P at the same time.

Moreover, the regions with fast economic growth, e.g., Kunshan, Wujiang, Changsu, and Tangcang, upstream of the Taihu Lake basin generally experienced an increase in soil C:N, C:P, and N:P between 1980 and 2000. The obvious increase in C:P in these regions was mainly due to a sharp decrease in soil P (Figure S2). Moreover, the increase in C:N in these regions mainly resulted from the decrease in soil N. The extent of the decrease in soil TP was larger than that of soil TN, resulting in the increase in soil N:P. In these developed areas with the rapid development of small- and medium-scale factories and tourist agriculture, the traditional agriculture that required large exogenous N and P inputs was reduced. As a consequence, soil C:N, C:P, and N:P experienced obvious increases.

As the spatial distribution of soil elemental stoichiometry was closely correlated with LUTs in different regions of the Taihu Lake basin, the changes in LUTs in these regions in the future will further shape the spatial distribution pattern.

### 4.4. The Effect of Agricultural Land Use Types on the Variation in Soil Elemental Stoichiometry

In the Taihu Lake basin, the variations in soil C:N, C:P, and N:P were mainly related to ALUTs and their corresponding management methods. Although a large area of agricultural land was transformed to construction land, the national soil surveys for soil C, N, and P were derived from agricultural lands, e.g., paddy fields, drylands, grassland, and woodlands. Paddy fields were the major ALUTs in this region, most of which experienced a substantial increase in soil C:N from the 1980s to the 2000s. Meanwhile, some paddy fields underwent a decrease in soil C:N. Normally, mechanical tillage was widely employed, which could accelerate SOC and N mineralization of agricultural land [50], thereby lowering C:N ratios as C losses as $CO_2$ have been expectedly higher than those for N. This trend can be even stronger as N fertilization or biological fixation adds more N to the system, which can also enhance SOC depletion by priming effects [51]. Similarly, low C:N ratios have been reported for soils under leguminous crops where N fixation is efficient and SOC decomposition is accelerated or in acidic soils [52]. However, an increase in the soil C:N of agricultural soil can occur when the supply of organic material (e.g., straw) substantially exceeds the increase in N fertilizer use. Similarly, most dry land area saw an increasing trend for soil C:N.

The changes in paddy fields and dry land were mainly transformed to each other or woodlands. Most of the transformed fields underwent an obvious increase in soil C:N, whereas some of them had a substantial decrease in soil C:N. This means that the agricultural soils generally experienced an increase in soil C:N after the rotation pattern changed. We speculate that the decrease in soil C:N was mainly from the alteration of paddy fields or drylands to economic forests that generally received excessive exogenous N input but without the simultaneous input of a large quantity of organic material (e.g.,

straw), as we discussed in the previous section [39]. The total woodland area experienced little change during 1980–2000. However, there was a contrasting trend for soil C:N changes between woodland areas that underwent alteration and remained unchanged. Among woodlands that had been transformed to other ALUTs, 68.0% of woodland soil had an obvious increase in soil C:N, as they were mainly transformed to paddy fields and dry land. In addition, most paddy fields, drylands soils, and woodlands, regardless of their LUT changes, experienced an increase in soil C:P and N:P. This is mainly because the exogenous input of organic materials and N fertilizer contributed to the increase in soil C:P and N:P, which exceeded the increase in P input.

## 5. Conclusions

The C:P (11:1~59:1) and N:P (1.2:1~7.6:1) ratios of agricultural soil in the Taihu Lake basin were far lower than the well-constrained values (C:N:P = 186:13:1) based on samples from forest, shrubland, and grassland at a global scale. The soil C:N was slightly lower than the well-constrained values, which were comparable to the values of the published average C:N ratios values across the world in the 1990s (C:P = 13:1~60:1, N:P = 0.9:1~5:1). Generally, the soil C:N and C:P ratios experienced consistent increases in the Taihu Lake basin from the 1980s to 2000 due partly to the continuous straw return practices. The regions with fast economic growth in the upstream of the Taihu Lake basin generally experienced increases in soil C:N, C:P, and N:P due to the reduction in agricultural land. The changes in LUTs and their corresponding management practices shaped the spatial and temporal characteristics of soil C:N:P in the Taihu Lake basin. The results suggested that the imbalanced C:N:P stoichiometry characteristics of agricultural soil may lead to decoupled C, N, and P biogeochemical cycles, thereby having an adverse impact on soil carbon sequestration.

**Supplementary Materials:** The following supporting information can be downloaded at: https://www.mdpi.com/article/10.3390/agriculture13020484/s1, Table S1: Agricultural land area changes in different land use type during the periods of 1980–2000 and 2000–2016 in Tai Lake basin; Figure S1. Temporal and spatial distribution of top soil C, N and P in 1980s and 2000s, respectively, in Taihu Lake basin; Figure S2. Changes of top soil C:N, C:P and N:P ratio between 1980s and 2000s in Taihu Lake basin. The expression unit of the changes is "%".

**Author Contributions:** Conceptualization, C.L. and Z.Y.; methodology, C.L. and Z.Y.; formal analysis, C.L. and Z.Y.; investigation, C.L. and Z.Y.; resources, Z.Y. and Y.D.; data curation, Y.D., C.L. and Z.Y.; writing—original draft preparation, C.L. and J.L.; writing—review and editing, P.L., J.L. and C.L.; supervision, Y.G. and P.L.; visualization, J.L.; project administration, W.S., Y.G. and J.L.; funding acquisition, W.S., Y.G. and J.L. All authors have read and agreed to the published version of the manuscript.

**Funding:** This research was jointly supported by the Jiangsu R & D Special Fund for Carbon Peaking and Carbon Neutrality (grant number: BK20220014), the National Natural Science Foundation of China (41571458, 41907026, 41871209), the China Postdoctoral Science Foundation (2018M632253), and the Open Foundation of Hebei Key Laboratory of Wetland Ecology and Conservation (hklk202005; hklz201904).

**Institutional Review Board Statement:** Not applicable.

**Data Availability Statement:** All data are fully available without restriction.

**Acknowledgments:** We extend a special thanks to Hongjie Di (College of Agriculture and Life Sciences, Lincoln University) for English editing and Yuhan Niu (Co-Innovation Center for Sustainable Forestry in Southern China, Nanjing Forestry University) for the literature download from the China National Knowledge Infrastructure (CNKI) database and ISI-Web of Science.

**Conflicts of Interest:** The authors declare no conflict of interest.

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
