# Peer review of "Temporal and Spatial Variations in Soil Elemental Stoichiometry Coupled with Alterations in Agricultural Land Use Types in the Taihu Lake Basin"

_agriculture, doi:10.3390/agriculture13020484_

Round 1

Reviewer 1 Report

This manuscript compared soil C:N:P stoichiometry under different agricultural land use types temporally and spatially in the Taihu Lake basin. This study is timely to provide valuable information for understanding the link between land use types and soil element stoichiometry. However, some concerns and syntax errors should be addressed before further consideration. 

L. 1 and 22. variations 

L. 25 stoichiometric

L. 29 significantly or not?

L. 63 change "of" to " between"

L. 70 Since the last century

L. 76 delete "of course"

L. 78 have resulted

L. 89 What are the kinds of relationships?

L. 106 delete "Therefore". Avoid using conjunctive adverbs and transitions at the beginning of a new paragraph. 

L. 106 objectives are

L. 113 This is too ambitious to uncover mechanisms in this study. 

L. 113-115 This sentence is confusing!

L. 225 delete "and"

L. 226 again, delete "and"

L. 227 What is the meaning of "with the LUTs remaining unchanged"

L. 279 delete the space between "paddy" and "had"

L. 285 delete "and"

L 320 Responses

L. 333 Please add p and r values for each correlation.

L. 354-359 What is your point?

L. 370 plant C input (above litter, rhizodeposits, and root mass) would be more appropriate.

L. 410 "increased" and "more" express the same meaning. Please choose one of them.

L. 413 It is unclear

L. 422 If you did not describe Figs 5 and 6 in the result section, you can move these two figures into the supplementary material. 

L. 431 meta-data

L. 483 What is the meaning of "or in aridic soils".

L. 500 regardless of their LUT changes

L. 508 "be comparable to" is so vague. Please specify it. 

Author Response

Please see the pdf file submitted.

Reviewer 2 Report

This is an interesting topic, linking changes in C:P:N to changes in land use over time.  It is also a difficult thing to do because of the many variables affecting these elements within different land use types.  Overall the paper is fairly well written but the presentation of results can be confusing at times and could perhaps be pared down to simplify it.  The discussion does a fairly good job of pointing out the many variables that could be contributing to changes over time but doesn't really describe the decoupling of the biogeochemical cycles that is cited in the conclusions and some cleaning up of the language would improve that section.

Author Response

Please see the pdf file submitted and responces have been listed  in the space of admin. 

Reviewer 3 Report

The manuscript agriculture-2155315 submitted by Chonggang Liu et al. and entitled "Temporal and spatial variation in soil elemental stoichiometry coupled with alterations in agricultural land use types in the Taihu Lake basin" presents an interesting scientific contribution related to the study of C:N:P stoichiometric in Taihu Lake basin due to the land use change.

In general, the manuscript is well written and the calculations made were correctly applied to obtain a reliable analysis of soil conditions and the effect of land use changes on C:N:P ratio.

Abstract: its quality is good enough

Keywords: must be revised; avoid repeating the same word already present in the title.

Introduction: Good; provides a good presentation of the topic, the need for knowledge on the subject and formulates hypotheses.

Materials and Methods: clear and well detailed

Results: well written, clear and well detailed and clearly understandable.

Discussion: covers every single aspect considered by the experimental activity by seeking a justification or rationale for each observed result. The level of detail is good.

Conclusions: based on the information retrieved and discussed above.

My specific comments, which I hope will help the authors to improve their manuscript, are enclosed in the attached file.

Author Response

Please see the pdf file submitted and the respinces have been listed in the sapce of admin.
